# Can women's reports in client exit interviews be used to measure and track progress of antenatal care services quality? Evidence from a facility assessment census in Malawi

**Martina Mchenga**[1,2]*, **Ronelle Burger**[3], **Dieter von Fintel**[3]

**1** Centre for Social Science Research, University of Cape Town, Cape Town, South Africa, **2** Graduate School of Economic Sciences and Management, Stellenbosch University, Stellenbosch, Western Cape, South Africa, **3** Economics Department, Stellenbosch University, Stellenbosch, South Africa

\* martinamchenga@gmail.com

## Abstract

### Introduction

Unlike household surveys, client exit interviews are conducted immediately after a consultation and therefore provides an opportunity to capture routine performance and level of service quality. This study examines the validity and reliability of women's reports on selected ANC interventions in exit interviews conducted in Malawi.

### Methods

Using data from the 2013–2014 Malawi service provision facility census, we compared women's reports in exit interviews regarding the contents of ANC received with reports obtained through direct observation by a trained healthcare professional. The validity of six indicators was tested using two measures: the area under the receiver operating characteristic curve (AUC), and the inflation factor (IF). Reliability of women's reports was measured using the Kappa coefficient (κ) and the prevalence-adjusted bias-adjusted kappa (PABAK). Finally, we examined whether reporting reliability varied significantly by individual and facility characteristics.

### Results

Of the six indicators, two concrete and observable measures had high reporting accuracy and met the validity criteria for both AUC $\geq$ 0.7 and 0.75>IF>1.25, namely whether the provider prescribed or gave malaria prophylaxis (AUC: 0.84, 95% CI: 0.83–0.86; IF: 0.96) or iron/folic tablets (AUC: (0.84 95% CI: 0.81–0.87; IF:1.00). Whereas four measures related to counselling had lower reporting accuracy: whether the provider offered counselling about nutrition in pregnancy (AUC: 0.69, 95%CI: 0.67–0.71; IF = 1.26), delivery preparation (AUC: 0.62, 95% CI: 0.60–065; IF = 0.99), pregnancy related complications (AUC: 0.59, 95%CI: 0.56–0.61; IF = 1.11), and iron/folic acid side effects (AUC:0.58, 95% CI: 0.55–0.60; IF = 1.42). Similarly, the observable measures had high reliability with both κ and PABAK values

**Data Availability Statement:** https://dhsprogram. com/methodology/survey/survey-display-424.cfm.

**Funding:** The authors received no specific funding for this work.

**Competing interests:** The authors have declared that no competing interests exist.

in the ranges of $\geq 0.61$ and $\geq 0.80$. Respondent's age, primiparous status, number of antenatal visits, and the type of health provider increased the likelihood of reporting reliability.

## Conclusion

In order to enhance the measurement of quality of ANC services, our study emphasizes the importance of carefully considering the type of information women are asked to recall and the timing of the interviews. While household survey programmes such as the demographic health survey and multiple indicator cluster survey are commonly used as data sources for measuring intervention coverage and quality, policy makers should complement such data with more reliable sources like routine data from health information systems.

## Introduction

In low and middle-income countries (LMICs), relatively high antenatal care (ANC) coverage still continues to co-exist with high maternal and neonatal mortality rates [1–3]. In Malawi, for example, 95% of women attend ANC at least once, while maternal mortality is estimated at 497 per 100,000 live births and neonatal mortality at 27 deaths per 1,000 live births [4]. This weak relationship between ANC use and maternal and new-born survival has motivated a shift in the focus from quantity of care to content and quality of care provided [3, 5]. This shift to focusing on quality of care underscores the need to monitor and track progress in the coverage and quality of care given [6]. Hence, accurate and high quality data on coverage and quality indicators is critical for global monitoring of trends and should be available at national levels to provide actionable information to achieve health goals [7].

In Malawi, similar to most LMICs household surveys such as the Demographic and Health Surveys (DHS) and the Multiple Indicator Cluster Surveys (MICS) are mostly used to track both coverage and quality of maternal and new-born services including antenatal care. However, a major limitation with household surveys is the long recall period (up to 5 years) [4]. Their reliability also depends on whether the woman is able to provide a consistent recollection of what happened in the past, and that a woman's current characteristics or conditions do not significantly influence her ability to accurately remember what happened (anchoring bias) [8]. For example, if a child had satisfactory birthweight, and is growing up well, mothers may be more likely to recall quality of care in a positive light, leading to upward bias in quality measurement. Moreover, evidence shows that women's recall of maternal and immediate postnatal interventions changes over time and that the longer the period, the less accurate the recall [7, 9], thereby questioning the validity of using household surveys with longer recall for monitoring and tracking quality of MNCH interventions.

Given the shortfalls of household surveys, facility level data provides a potentially reliable alternative. There are a number of methods that are used to collect service coverage and quality of care information at the facility level including review of medical records [10]; direct observation of clinical consultations by an expert; client exit interviews [11]; and mystery patients [12]. Of these, medical records have been shown to be the most reliable method of collecting data on service coverage and quality of health care [10, 13] as they allow retrospective assessment of routine provider performance. However, in LMICs including in Malawi, the use of medical records is often of little use due to incomplete, inconsistent or even non-existent records, particularly at public facilities [14]. As such, service coverage or quality of care estimates at facility level, are better collected using direct observation or client exit interviews.

Direct observation involves recordings of the provider's actions during a consultation by an independent observer. The approach is mostly considered a gold standard, but is costly to implement [15]. Franco et al. [16] argued that information derived from direct observation, when the independent observer simultaneously records the providers actions using structured checklists to assess whether he/she is following a set of guidelines, has the potential to provide one of the most complete and reliable pictures of what providers do. Unlike household surveys, client exit interviews are conducted immediately after a consultation and therefore provides an opportunity to capture routine performance and level of service quality [17]. Currently, limited research exist that assess the client's ability to accurately recall in exit interviews the services they received and the quality during a consultation. Exit interviews—just like household interviews—are less costly than direct observations. However, they provide a better alternative of collecting data given their shorter recall period than household surveys. Exit interviews therefore strike a balance between limiting costs and obtaining immediate patient recall, and it is therefore important to understand whether they provide useful information in LMIC health systems.

Existing evidence shows that women are more likely to report with accuracy on concrete and observable measures including weight and height measurement compared to measures related to counselling. For example, results from a multicountry study which includes Ecuador, Uganda and Zimbabwe shows that for the majority of indicators, agreement was good to excellent and was relatively high on the indicators of interpersonal relationships but lower on those related to counselling [18]. A similar finding was also reported in a study conducted in Bangladesh, Cambodia and Kenya showing that women's reports in exit interviews of antenatal and postnatal care received had higher validity for indicators related to concrete and observable actions, as opposed to information or advice given [19]. Another multicountry study conducted in Malawi, Haiti and Senegal found overall low agreement in counselling topics related to ANC [20]. As far as we know, this is the only study that includes Malawi, however, the study only focuses on counselling components of maternal and child health services. Our study provides an extension to this study by focusing on ANC interventions in Malawi and includes other aspects of care besides counselling.

Given the renewed focus on measurement of quality- adjusted coverage, as well as the limited number of studies that have sought to assess self-reported ANC interventions, additional validation work is warranted. The present study aims to extend research findings to date by assessing the validity of a set of antenatal indicators that reflect a range of recommended interventions and counselling procedures in Malawi. Using heterogeneity chi-squared tests, the study also examines whether reporting reliability varies significantly by individual and facility level characteristics.

## Data and methods

### Study design

This study was a secondary analysis of de-identified data from the facility assessment survey in Malawi, which is readily available on the Demographic Health Survey program website upon request (https://dhsprogram.com/Data/). Original survey participants offered their informed consent to the DHS survey enumerators.

The 2013–14 Malawi Service Provision Assessment (MSPA) was designed to be a census of all formal-sector health facilities in Malawi with a master list of 1,060 facilities provided by the Central Monitoring and Evaluation Division (CMED) of the Malawi Ministry of Health. Of the 1,060 formal health facilities that were visited during the assessment, 83 facilities were permanently closed, unreachable, duplicates of other facilities, or refused to participate [21]. Data is therefore available for a total of 977 facilities.

The survey used four questionnaires to collect data on various aspects of quality of care. These include: (1) an inventory questionnaire examined the availability of services and features of the facility; (2) the health worker interview collected information from 8–15 selected health workers on their duties, training, and demographic characteristics; (3) observation protocols were recorded by a health worker with expert knowledge and experience to assess providers' adherence to clinical guidelines during consultations. The number of observations ranged between 5–15 consultations per provider and per service depending on the size of the facility. Facilities where direct observations were conducted were chosen at random to avoid sample selection bias. In addition four exit interviews were conducted at each facility. The clients whose ANC visits were observed were asked questions about their ANC visit and demographic information. This analysis compares reports from observation protocols by healthcare professionals and exit interviews.

## Sample size

The total number of facilities where the observational protocol was administered was 412. A total of 2105 women were available on the day of assessment at the selected observed facilities, however, 2068 agreed to be both observed during the consultation and interviewed after the consultation, representing a 98% response rate. In ANC consultations, experts observed whether health workers conducted routine tests and prevention procedures outlined in ANC guidelines. In the exit interviews, women were asked to recall about the services they received, their perception about the services, as well as other socio-demographic characteristics such as education level, number of ANC visit and whether it was their first pregnancy, among other factors.

## Variables of interest

Indicators included in this study were selected based on two criteria: (1) interventions that are outlined in both the WHO ANC guidelines [22] and Malawi ANC guidelines [23] and (2) availability of the indicators in both the observation guide and the exit interview questionnaires. We identified the following six indicators in both questionnaires; (i) whether providers prescribed or gave fansidar for malaria prevention; (ii) whether providers prescribed or gave iron and folic tablets; (iii) whether providers explained the side effects of iron and folic tablets; (iv) whether providers discussed the importance of good nutrition during pregnancy; (v) whether providers discussed delivery preparation and; (vi) whether providers discussed pregnancy related complications.

Binary variables were generated to determine whether women received any of the mentioned ANC elements using the two data sources. In the case of direct observation, the indicator for the provision of an ANC service was coded as 1 if the observer noted that the service was provided by the provider, and 0 otherwise. When the provider was not observed providing the service or cases with missing information were recorded as 0 and considered not to have taken place [24]. Similarly, during client exit interviews, women were asked about their receipt of specific ANC services in one of the following contexts; (1) during the current observed visit; (2) both the current observed visit and the past visit; or (3) the last visit only. To ensure comparability with direct observation, client exit indicators were coded as 1 if only a woman reported receiving any of the ANC services either during the current visit or during both the current and previous visits.

## Analytical methods

The study applied validity and reliability measures to assess the client's ability to accurately recall in exit interviews the services they received and the quality during a consultation. In the

case of validity (defined in this context as the extent to which client exit reports about the services they received align with direct observation reports), we constructed a two by two table to calculate sensitivity (the true positive rate) and specificity (true negative rate) for each indicator. Using both sensitivity and specificity, we estimated the area under the receiver operating characteristic curve (AUC) and its corresponding 95% CI following a binomial distribution [25]. Based on the literature (19,26), specificity and sensitivity are usually interpreted as follows: high ($\geq$90% and 100%); good ($\geq$80 and < 90%) and low (<80%). Meanwhile, the AUC can be interpreted as 'the average sensitivity across all possible specificities [25]. An AUC of 0.5 indicates an uninformative test and an AUC of 1.0 represents perfect accuracy (100% sensitivity and 100% specificity) [25].

To assess population-level validity for each indicator, we calculated the degree to which an indicator would be overestimated or underestimated in the exit interviews using the inflation factor (IF). Specifically, the IF is the ratio of the indicator's estimated population-based survey prevalence to the indicator's 'true' (observed) prevalence [26]. The population-based survey prevalence (Pr) is estimated as follows;

$$Pr = P*(SE + SP - 1) + (1 - SP)$$

Where SE is sensitivity and SP is the specificity and P is the indicator's true observed prevalence. We used an IF cut-off between 0.75 and 1.25 as the benchmark for low population-level bias [26]. Following McCarthy et al. [19], indicators considered to have both high individual accuracy with an estimated value of AUC equal to 0.70 or higher and low population-level bias (0.75<IF<1.25) were considered to have acceptable validity.

For validity studies, adequate sample size is important to ensure precise estimates of specificity and sensitivity. In our study, given that the assessed indicators were health promoting and prevention focused and should be nearly universal, we anticipated the prevalence of indicators to range from 50% to 80% as per McCarthy et al. [19]. Since the interviews were conducted on the same day of the consultation and recall was immediate, we assumed levels of moderate to high sensitivity (60%-70%) and specificity (70%–80%). Using Buderer's formula [27], a sample size of 400 was considered sufficient for the anticipated sensitivity and specificity levels with at least 7% precision. Therefore, a sample of 2068 is more than enough to give us precise validity estimates.

On the other hand, reliability measures or agreement between direct observation and women's reports in the exit interviews were calculated using the Kappa coefficient (κ) with corresponding 95% confidence interviews (95% CI). The Kappa coefficient (κ) is the proportion of agreement beyond expected chance agreement [28]. Although Kappa remains the most widely used measure of agreement, several authors have pointed out data patterns that produce a κ with paradoxical results [28]. Therefore, the prevalence adjusted bias adjusted Kappa (PABAK) measure will also be reported. PABAK gives the true proportion of agreement beyond expected chance agreement regardless of unbalanced data patterns [28]. Both κ and PABAK values were interpreted using the Landis and Koch categorization [29] as follows: almost perfect (>0.80), substantial (0.61–0.80), moderate (0.41–0.60), fair (0.21–0.40), slight (0.00–0.20) and poor (<0.00).

The study further examines whether women's reporting reliability varied significantly by their sociodemographic characteristics or facility characteristics. To create the agreement variable, we developed a composite score of ANC indicators using only indicators that had adjusted PABAK coefficients in the categories of almost perfect to moderate [30]. In this study, four out of the six indicators were in the categories of almost perfect to moderate as shown in Table 4. We assigned a value of 1 to each indicator agreement (1 = yes/yes or no/no) and 0

otherwise. We multiplied the score by 25 to get a range of values between 0–100. The score was dichotomized into low ($\leq$70 points as 0) and high agreement ($\geq$ 75 points as 1) which was then used as the dependent variable. The agreement variable was then stratified by covariates hypothesized to influence women's reporting ability. These were: educational attainment, age of the client, number of pregnancies, number of visits to the facility during the current pregnancy, region of residence (north, central or south), type of facility (hospital, health center or dispensary), facility managing authority (private or public) and type of provider (doctor, clinician, nurse or midwife). Significance was determined using the heterogeneity chi-squared tests. All analyses were conducted in Stata 17.

## Results

### Sample characteristics of the client exit interviews

Table 1 provides characteristics of the women that were available on the day of assessment and had agreed to be both observed and interviewed. Most of the women were 25 years or older (45%). About 14% of the women never attended school, with the majority (61%) reporting to have at least primary education. About 24% of the women were interviewed during their first pregnancy. Of the women who came to the facility for ANC, 42% visited that particular facility their first time to receive ANC services. The majority of the women (57%) visited health centers for their ANC visits, and of the facilities they visited, 74% were public facilities. The majority of the ANC consultations were conducted by midwives (75.41%), followed by nurses (20.60%). Only about 8% of the providers used visual aids in the ANC consultations.

### Characteristics of health facilities where direct observations were conducted

In Table 2 we present the characteristics of the facilities where ANC direct observations and client exit interviews were conducted. About 72% of the facilities were health centers and rural based (82%). Almost 67% of the facilities were government owned. Most of the facilities were located in the southern region (44%) and the central region (41%).

### Validation measures

Fewer than 1% of the women responded to the questions with an answer of "don't know", limiting the extent of missing information (Table 3). The validation estimates show significant variations in specificity and sensitivity across the indicators. The lowest sensitivity (0.33, 95% CI 0.31–0.35) was observed for whether the woman was informed about the side effects of iron tablets. Whereas the lowest specificity was observed for whether the woman was informed about delivery preparation (0.44, 95%CI 0.42–0.46). In general, the sensitivity of the self-reported exit interviews was high ($\geq$90%) for 16.67% of the indicators evaluated, good ($\geq$80 and $<$ 90%) for 33.33% and low ($<$80%) for 50% of the indicators (all of them in the domain of "counselling"). Specificity was low ($<$80%) for 66.67% of indicators and high for whether the provider prescribed malaria prophylaxis (84%, 95% CI: 0.82–0.86) and whether the woman was informed of iron tablets side effects (89%, 95% CI: 0.87–0.90).

Four indicators had AUC results of 0.60 or greater (Table 3 and Fig 1). The most accurately reported responses were to the question on whether the provider gave or prescribed malaria prophylaxis (0.84, 95% CI: 0.83–0.86); whether the provider gave or prescribed iron tablets (0.84 95% CI: 0.81–0.87); whether the provider discussed the importance of good nutrition in pregnancy (0.69, 95%CI: 0.67–0.71) and whether the woman was informed of delivery preparation (0.62, 95% CI: 0.60–065).

**Table 1. Selected social and demographic characteristics of the analysis sample.**

| Variables | Frequency | Percent (%) | 95% CI |
|---|---|---|---|
| **Age at last birthday** | | | |
| 13–19 | 405 | 20.26 | (18.41 22.24) |
| 20–24 | 659 | 32.95 | (30.67 35.32) |
| 25+ | 936 | 46.79 | (44.29 49.31) |
| **Mother's education level** | | | |
| Never attended school | 289 | 13.95 | (12.33 15.75) |
| Primary | 1,268 | 61.31 | (58.86 63.70) |
| Secondary or higher | 512 | 24.74 | (19.88 24.10) |
| **Parity** | | | |
| First pregnancy | 505 | 24.40 | (22.38 26.54) |
| Not first pregnancy | 1,563 | 75.60 | (73.38 77.10) |
| **Number of visits for the pregnancy to the facility** | | | |
| First visit | 873 | 42.22 | (39.81 44.66) |
| Second visit | 496 | 24.01 | (21.96 26.18) |
| 3+ visits | 699 | 33.78 | (31.51 36.12) |
| **Type of facility** | | | |
| Hospital | 828 | 40.05 | (37.47 42.69) |
| Health Center (including maternity) | 1,183 | 57.18 | (54.59 59.73) |
| Dispensary/Clinic/Health post | 57 | 2.77 | (2.19 3.50) |
| **Facility managing authority** | | | |
| Public | 545 | 73.65 | (71.59 75.62) |
| Private | 1,523 | 26.35 | (24.38 28.41) |
| **Type of provider** | | | |
| Doctor | 10 | 0.50 | (0.27 0.92) |
| Clinician | 72 | 3.49 | (2.84 4.29) |
| Nurse | 426 | 20.60 | (18.44 22.93) |
| Nurse Midwife | 1,560 | 75.41 | (73.05 77.63) |
| **Provider used visual aids during the consultation** | | | |
| Yes | 164 | 7.95 | (6.83 9.22) |
| No | 1,897 | 92.05 | (89.92 92.37) |

Source: Own derived from the 2013/2014 Service Provision Assessment surveys.

The other criterion of acceptable validity was an inflation factor between 0.75 and 1.25, and four indicators met this criterion. These were: the woman was given or prescribed malaria tablets (0.96); was given or prescribed iron tablets (1); the woman was informed of what she needed to do to prepare for delivery (0.99); and the woman was informed of pregnancy related complications (1.11). Two of the indicators with an inflation factor of greater than 1.25 had lower observed prevalence rates in comparison to the other indicators. For example, only 15% of the women, were observed being informed about iron tablets side effects (IF = 1.42) and 39% of women were observed being informed about proper nutrition or important foods during pregnancy (IF = 1.26).

## Measures of agreement

Table 4, shows Kappa estimates. Of the six indicators we observed slight agreement strength with expert direct observation for two indicators (κ between 0.00–0.20), fair agreement for two

**Table 2. Characteristics of facilities.**

| Characteristics | Frequency | Percent (%) | 95% CI |
|---|---|---|---|
| **Type of facility** | | | |
| Hospital | 89 | 21.65 | (17.93 25.89) |
| Health Center (including maternity) | 295 | 71.67 | (67.10 75.84) |
| Dispensary/Clinic/Health post | 28 | 6.68 | (4.58 9.64) |
| **Managing Authority** | | | |
| Public/government | 275 | 66.69 | (61.97 71.10) |
| Private | 137 | 33.31 | (28.90 38.03) |
| **Location of facilities** | | | |
| Rural | 339 | 82.28 | (78.27 85.69) |
| Urban | 73 | 17.72 | (14.31 21.73) |
| **Region** | | | |
| North | 62 | 15,19 | (12.01 19.02) |
| Central | 170 | 41.21 | (36.53 46.05) |
| South | 180 | 43.61 | (38.87 48.46) |

Source: Own based on 2013/2014 SPA surveys

indicators (κ between 0.21–0.40), and substantial agreement for two indicators (κ > = 0.60). The agreement strength using PABAK was higher than that obtained with the Kappa coefficient in three (50%) indicators; whether the woman was prescribed or given iron tablets, whether the woman was informed of the side effects of iron tablets, and whether the woman was informed of what to prepare for delivery. The other three indicators were similarly categorised by PABAK and Kappa values.

**Table 3. Sensitivity and specificity of reporting in exit interviews compared to direct observation in service provision assessment survey.**

| Variable | Client reported (n) | Don't know (n) | N | Client reported prevalence | Observed prevalence | Estimated population prevalence based on sensitivity & specificity | Sensitivity (95% CI) | Specificity (95% CI) | AUC | IF | Met both criteria |
|---|---|---|---|---|---|---|---|---|---|---|---|
| Given or prescribed malaria prophylaxis | 2,068 | 0 | 2,068 | 59.61% | 62% | 59% | 0.86 (0.85; 0.88) | 0.84 (0.82; 0.86) | 0.84 (0.83; 0.86) | 0.96 | Y |
| Given or prescribed iron tablets | 2,068 | 0 | 2,068 | 86.49% | 86.50% | 87.29% | 0.96 (0.95; 0.97) | 0.71 (0.69; 0.73) | 0.84 (0.81; 0.87) | 1 | Y |
| provider ever discussed side effects of iron tablets | 2,050 | 10 | 2,068 | 13.59% | 9.52% | 13.49% | 0.33 (0.31; 0.35) | 0.89 (0.87; 0.90) | 0.58 (0.55; 0.60) | 1.42 | N |
| provider ever discussed diet/nutrition during pregnancy | 2,061 | 7 | 2,068 | 49.89% | 39.14% | 48% | 0.74 (0.72; 0.76) | 0.66 (0.64; 0.69) | 0.69 (0.67; 0.71) | 1.26 | N |
| provider ever discussed preparation for delivery | 2,061 | 7 | 2,068 | 74.86% | 74.63% | 74.63% | 0.80 (0.79; 0.82) | 0.44 (0.42; 0.46) | 0.62 (0.60; 0.65) | 0.99 | N |
| provider talked about danger signs in pregnancy | 2,062 | 6 | 2,068 | 51.36% | 46.30% | 51.51% | 0.61 (0.59; 0.63) | 0.56 (0.54; 0.59) | 0.59 (0.56; 0.61) | 1.11 | N |

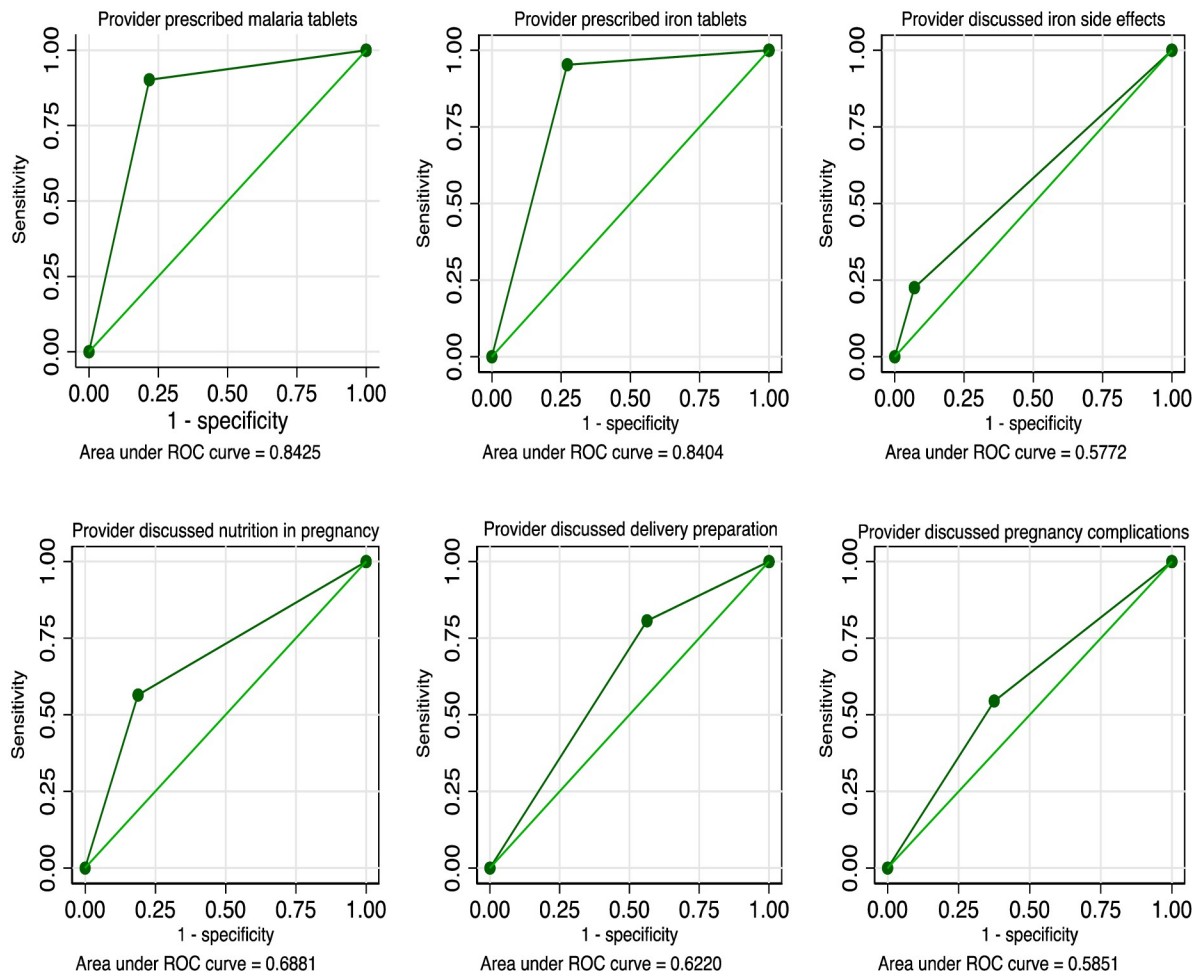

**Fig 1. ROC curves for comparison between direct observation reports and women's self-reports in exit interviews. Source**: Authors from 2013–2014 Malawi Service Provision Assessment (MSPA).

### Reliability of women's self-reports by covariates

In Table 5 we present results for the bivariate analysis and we found that agreement was likely to be high among pregnant women who were younger than 25 years, with primiparous status, those who were on their first ANC visit, those whose health care provider was not a doctor and those whose facilities were located in the central/southern region.

## Discussion

Providing high quality maternal health care services is essential for improving the health and survival of women and new-borns. To effectively monitor progress in delivering quality maternal health care services, accurate information regarding the services received is indispensable. This study builds upon previous validation studies focused on maternal and new-born care interventions. Specifically, it investigates the validity and reliability of women's self-reports obtained through client exit interviews, focusing on the specific components of antenatal care received during consultations. By comparing these self-reports to direct observation reports

**Table 4. Concordance measures of ANC indicators from the direct observations and client exit interviews.**

| Indicator | Concordance (%) | | Discordance (%) | | % Agreement (95% CI) | Kappa (95% CI) | P-Value | Agreement strength | PABAK (95% CI) | Agreement strength | P-Value |
|---|---|---|---|---|---|---|---|---|---|---|---|
| | Y/Y | N/N | Y/N | N/Y | | | | | | | |
| Given or prescribed malaria prophylaxis | 86.81 | 84.90 | 13.19 | 15.10 | 86.06 (84.59 87.58) | 0.71 (0.68 0.74) | 0.0000 | Substantial | 0.72 (0.69 0.75) | Substantial | 0.0000 |
| Given or prescribed iron tablets | 95.84 | 73.39 | 26.61 | 4.16 | 92.81 (91.69 93.92) | 0.69 (0.65 0.74) | 0.0000 | Substantial | 0.86 (0.83 0.88) | Almost perfect | 0.0000 |
| provider ever discussed side effects of iron tablets | 32.07 | 88.36 | 11.64 | 67.93 | 82.98 (81.33 84.64) | 0.17 (0.11 0.23) | 0.0000 | Slight | 0.66 (0.63 0.69) | Substantial | 0.0000 |
| provider ever discussed diet/ nutrition during pregnancy | 74.85 | 66.26 | 33.74 | 25.15 | 69.63 (67.64 71.63) | 0.39 (0.35 0.43) | 0.0000 | Fair | 0.39 (0.35 0.43) | Fair | 0.0000 |
| provider ever discussed preparation for delivery | 81.28 | 44.11 | 55.89 | 18.72 | 71.88 (69.92 73.83) | 0.26 (0.21 0.30) | 0.0000 | Fair | 0.44 (0.40 0.48) | Moderate | 0.0000 |
| provider talked about danger signs in pregnancy | 60.43 | 56.45 | 43.55 | 39.57 | 58.29 (56.16 60.43) | 0.17 (0.13 0.21) | 0.0000 | Slight | 0.17 (0.12 0.21) | Slight | 0.0000 |

**Table 5. Women's reporting reliability by individual and facility factors.**

| Characteristics | Low agreement (%) | High agreement (%) | Chi-square P-Value |
|---|---|---|---|
| **Age category** | | | |
| 13–19 | 10.48 | 89.52 | |
| 20–24 | 12.73 | 87.27 | 0.010** |
| 25+ | 16.31 | 83.69 | |
| **Maternal education** | | | |
| Never attended school | 17.02 | 82.98 | |
| Primary | 13.27 | 86.73 | 0.257 |
| Secondary or higher | 13.98 | 86.02 | |
| **Parity** | | | |
| First pregnancy | 11.35 | 88.65 | 0.053* |
| ≥ 2 children | 14.80 | 85.20 | |
| **Number of visits for the pregnancy to the facility** | | | |
| First visit | 8.44 | 91.56 | |
| Second visit | 17.89 | 82.11 | 0.000*** |
| 3+ visits | 17.91 | 82.09 | |
| **Region** | | | |
| North | 17.72 | 82.28 | |
| Central | 15.14 | 84.86 | 0.010** |
| South | 11.49 | 88.51 | |
| **Type of facility** | | | |
| Hospital | 14.53 | 85.47 | |
| Health Center | 13.69 | 86.31 | 0.879 |
| Dispensary/Clinic/Health post | 14.58 | 85.42 | |
| **Facility managing authority** | | | |
| Public | 14.04 | 85.96 | 0.848 |
| Private | 13.72 | 86.28 | |
| **Type of provider** | | | |
| Doctor | 50.00 | 50.00 | |
| Clinician | 17.02 | 82.98 | 0.008*** |
| Nurse | 14.01 | 85.99 | |
| Nurse Midwife | 13.54 | 86.46 | |

conducted by an independent expert, the study aims to assess the consistency and accuracy of the information provided by the women.

In terms of validity measures, the findings indicate were more accurately reported in exit interviews when indicators related to concrete and observable interventions. For instance, indicators related to medical prescription and physical care such as whether the provider prescribed/gave malaria tablets or iron tablets met both the validation criteria. Both had an AUC above the 0.70 benchmark. Conversely, indicators related to more abstract concepts, particularly those pertaining to counselling or advice given, were less reliably reported. These specific indicators, such as recalling whether counselling was provided on side effects of iron tablets, diet/nutrition during pregnancy, preparation for delivery, and danger signs in pregnancy, did not meet the validation criteria. All of these indicators had an AUC below the benchmark of 0.70. These findings support existing evidence highlighting that women tend to report concrete and observable measures, such as weight and height measurements, or being given medical prescriptions more accurately than subjective measures like counselling [15, 18, 19, 30].

The greater accuracy in recalling physical aspects of care compared to being given advice or information has mixed support in prior validation studies. A recall study on postnatal care in Kenya and eSwatini found higher reporting accuracy for specific indicators of physical examination, while also noting relatively better recall for counselling indicators, including discussions on danger signs for the mother [31]. However, in this study, the indicator related to danger signs did not meet the validation criteria. The discrepancy between the present study and McCarthy et al. [31] may be attributable to differences in the stages of the pregnancy (antenatal vs postnatal) and the inclusion of observable measures in the study (medical prescription vs physical examination).

Similar to validity measures, indicators related to observable interventions, such as drug prescriptions, exhibited moderate to high agreement. This suggests that assessing quality of ANC services through women's self- reports in client exit interviews is reliable and comparable to using direct observation for those specific components. These findings align with a study conducted in Brazil, where antenatal clinic exams, including weight measurements, symphysis-fundal height, and blood pressure, reached almost perfect agreement [32]. Similarly, a multicountry study conducted in Ecuador, Uganda and Zimbabwe found that for the majority of indicators, agreement was good to excellent and was relatively high on the indicators of interpersonal relationships but lower on those related to counselling [18]. The ability of the patient to identify the reason for the ANC procedure or action recorded by the health professional during the ANC visit may contribute to these findings [32]. For example, the client may understand the reasons behind taking malaria tablets and/or iron tablets during pregnancy, which enhances their ability to accurately report on these interventions.

In general, the prevalence of counselling indicators was found to be relatively low in this study, even when assessed through direct observation. This finding is consistent with previous evidence from studies conducted in Lao [33] and a multi-country investigation in Burkina Faso, Uganda, and Tanzania [34]. These studies indicate that very few health workers provide adequate health education to pregnant women, and even among those who do, the information provided is often insufficient, particularly concerning pregnancy danger signs, nutrition, and other related topics [33, 34]. These findings raise significant concerns about the quality of antenatal counselling received by women, the methods used during counselling sessions, and the extent to which women comprehend and retain the information provided [9].

An analysis of assessment data from Haiti, Malawi, and Senegal collected between 2012 and 2014 found that the strongest predictor of ANC knowledge among clients was when there was an agreement between client and observer reports that counselling had been provided [20]. The low agreement between clients and observers regarding the occurrence of counselling found in this study potentially suggests that poor-quality counselling may have influenced clients' acknowledgment that counselling occurred [9]. Existing evidence shows that lack of communication training and unavailability of information education and communication (IEC) materials in public facilities are key reasons for poor quality counselling in low income countries [34]. Interestingly, in our study, two counselling indicators exhibited higher sensitivity compared to others: whether the provider discussed diet and nutrition during pregnancy as well as whether the provider discussed preparation for delivery. In these cases, it is possible that counselling was accompanied by observable actions (ICE material), such as visual aids demonstrating food groups or the use of recommended materials during delivery in public facilities, such as plastic black paper. In this context, improving health worker communication skills through training, for example, can enhance their ability to provide comprehensive health information and increase women's understanding of the care given.

Individual sociodemographic characteristics as well as facility characteristics may contribute towards the strength of agreement between data sources [32]. Distinguishing results by

covariates shows that at an individual level, women who were younger than 25 years, with primiparous status and those whose ANC visit was the first were likely to have higher agreement. A possible explanation could be linked to their perceived inexperience: women in these categories are likely to compensate for their uncertainty by being more attentive and seeking clarification where possible. On the other hand, women who are older or have other children and may believe that their prior experience sufficiently informed them about the pregnancy, and they may therefore be inattentive to new information given by the provider [32]. A study by Morón-Duarte et al. [32] reports similar findings, whereby higher maternal age and having previously given birth to ≥2 children were associated with a lower probability of high agreement between the antenatal card and the self-reported questionnaire.

Furthermore, our research revealed that women who received consultations from doctors displayed lower agreement levels compared to those who received antenatal care (ANC) consultations from clinicians, nurses, or midwives. This finding aligns with a Cochrane systematic review which demonstrated that care provided by nurses, in contrast to care provided by doctors, likely produces similar or improved health outcomes across a wide range of patient conditions [35]. Moreover, we also observed a notable level of agreement in healthcare facilities situated in the central/southern region. This observation leads us to believe that language barriers may be a contributing factor to low quality care. In the northern region of Malawi, the official language is Tumbuka, whereas Chichewa is less prevalent than in the southern and central regions. It is plausible that the language barrier between the healthcare provider and the client, particularly when the provider is not a native speaker of Tumbuka, may have contributed to the reduced agreement in the Northern region. A metareview study conducted in 2014 aligns with this finding and highlighted language barriers in information and communication as one of the primary obstacles to quality maternal, new-borns, and child healthcare [36]. Therefore, it is crucial to delve deeper into the issue of language and assess its impact on comprehension and information retention during clinical consultations.

### Limitations

The major strength of this study is the use of facility level census data and the use of direct observations as a gold standard. On the other hand, the interpretation of these results should be done with caution. First, our analysis was limited to only indicators which were available in both direct observation guide and client exit interview questionnaire. These indicators cannot comprehensively describe the ANC process. Second, while direct observation by an expert is considered to be the gold standard, it may also be imperfect. Differences in observer training protocols, facility practices or how apparent it was that a given intervention was implemented (especially the counselling components), among other factors, may contribute to differences in observer ratings across settings [19]. Last but not least, unlike regression analysis, the covariate analysis does not quantify the influence of the covariates on the reliability of women's self-reports.

In addition, relevant household factors such as wealth were are not typically recorded in exit interviews and direct observations. This limits the extent to which regression models can adequately control for contextual factors that could determine the extent of reporting agreement. Results should therefore be interpreted with caution. To overcome the stated limitations, future validation studies could potentially supplement quantitative findings with qualitative research to inform how women understand questions and specific terminology. It would also be interesting to examine how different approaches to women of different ages can affect the result of data collection and then agreement.

## Conclusion

The aim of the study was to investigate the potential reliability and accuracy of women's self-reports in assessing the quality of care obtained during antenatal care obtained through client exit interviews. To evaluate this, a comparison was made between the women's reports and direct observation reports conducted by an expert. The study utilized the 2013–2014 Malawi Service Provision Assessment (MSPA), which includes information from both expert direct observation and client exit interviews. Our findings indicate that women are capable of providing accurate and reliable reports on certain aspects of care received during antenatal care visits, particularly those pertaining to concrete and observable measures such a medical prescriptions. However, the study also reveals limited reliability and accuracy in subjective measures, such as counselling.

An important consideration of the relevance of these study findings for national and global monitoring efforts is that this study assessed women's immediate recall accuracy (at facility discharge). Results may therefore not directly generalise to the DHS and MICS that typically ask women to recall events related to a birth in the 2–5 years prior. Nonetheless, the findings provide valuable insights for measurement and monitoring of quality in the provision of maternal health services. While household survey programmes such as the DHS and MICS are commonly used as data sources for measuring intervention coverage and quality, policy makers should complement such data with more reliable sources like routine data from health information systems, otherwise there is a risk of making ill-informed policy decisions based on inaccurate information due to longer recall.

Furthermore, as new indicators, particularly those related to counselling, are proposed, they should undergo validity tests using various recall periods. Based on the results of such studies, data quality on antenatal care and maternal health services in general can be enhanced by limiting detailed and counselling related survey questions to births that occurred within the year preceding the survey.

## Author Contributions

**Conceptualization:** Martina Mchenga, Ronelle Burger, Dieter von Fintel.

**Data curation:** Martina Mchenga.

**Formal analysis:** Martina Mchenga.

**Methodology:** Martina Mchenga, Dieter von Fintel.

**Supervision:** Ronelle Burger, Dieter von Fintel.

**Validation:** Ronelle Burger, Dieter von Fintel.

**Writing – original draft:** Martina Mchenga.

**Writing – review & editing:** Martina Mchenga, Ronelle Burger, Dieter von Fintel.

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
