## [Decision Letter · Decision Letter 0]

10 May 2023

PONE-D-22-23821Can women’s reports in client exit interviews be used to measure and track progress of antenatal care quality? Evidence from a facility assessment census in MalawiPLOS ONE

Dear Dr. Mchenga,

Thank you for submitting your manuscript to PLOS ONE. After careful consideration, we feel that it has merit but does not fully meet PLOS ONE’s publication criteria as it currently stands. Therefore, we invite you to submit a revised version of the manuscript that addresses the points raised during the review process.

I would like to sincerely apologise for the delay you have incurred with your submission. It has been exceptionally difficult to secure reviewers to evaluate your study. We have now received three completed reviews; the comments are available below. The reviewers have raised significant scientific concerns about the study that need to be addressed in a revision.

Please revise the manuscript to address all the reviewer's comments in a point-by-point response in order to ensure it is meeting the journal's publication criteria. Please note that the revised manuscript will need to undergo further review, we thus cannot at this point anticipate the outcome of the evaluation process.

We look forward to receiving your revised manuscript.

Kind regards,

Miquel Vall-llosera Camps

Senior Editor

PLOS ONE

2. Please amend your manuscript to include your abstract after the title page.

Reviewers' comments:

Reviewer's Responses to Questions

**Comments to the Author**

1. Is the manuscript technically sound, and do the data support the conclusions?

Reviewer #1: Yes

Reviewer #2: Partly

Reviewer #3: Yes

2. Has the statistical analysis been performed appropriately and rigorously? 

Reviewer #1: Yes

Reviewer #2: Yes

Reviewer #3: Yes

3. Have the authors made all data underlying the findings in their manuscript fully available?

Reviewer #1: No

Reviewer #2: Yes

Reviewer #3: Yes

4. Is the manuscript presented in an intelligible fashion and written in standard English?

Reviewer #1: Yes

Reviewer #2: Yes

Reviewer #3: Yes

5. Review Comments to the Author

Reviewer #1: Overall, the authors addressed an essential topic: measuring prenatal care coverage and quality in Malawi. They conducted a descriptive study of the ANC quality indicators obtained by direct observation and women's self-report in existing interviews using data from the 2013-14 Malawi Service Provision Assessment. The introduction is concisely written and lays out the aims, objectives and rationale of the study clearly. The authors did an excellent job at elaborating on the materials and methods section and have provided sufficient detail to allow understanding and replication of the study. The results were interpreted well using different methods and clearly segregated into relevant parts. The discussion is also well written, and cites from other available evidence to put forward a strong argument in the favor of using other data sources than simply DHS/MICS to measure coverage. However, the authors do leave the room to elaborate more on implications of these findings in real life and for policymakers. They must add a few lines in the conclusion and hopefully sprinkle it throughout the discussion as well, that would solutions do they propose based on these findings. The authors point a valid problem but it would be ideal if they gave a clear answer to it as well; if not DHS, then what data source can be relied on in most scenarios to estimate coverage, given the resource limitations in LMICs?

Lastly, the authors have addressed all the comments made by previous reviewers, to the extent of changing their analytical approach and rewriting this paper. This is a commendable effort and should be rewarded by publishing with minor revisions (regarding addition of clear policy implications and proposed solutions).

Reviewer #2: Dear editor,

Thank you so much for inviting me to assess the scientific quality of the paper titled: “Can women’s reports in client exit interviews be used to measure and track progress of antenatal care quality? Evidence from a facility assessment census in Malawi” Manuscript Number: PONE-D-22-23821

Dear authors,

Thank you for addressing an important area maternal health. I suggest you to clarify the following points in your manuscript.

1. It is not good to call a researchers’ name when using their article. Example Franco et al. on page 3, Bessinger and BertrandBetter, McCarthy et al, Assaf et al on page 3, and throughout the discussion part. Better to say, “evidence shows, article indicates,….

2. Data and methods: The score was dichotomized into low (≤70 points as 0) and high agreement (≥ 75 points as 1) which was then used to conduct bivariate analysis using the score as the dependent variable on page 6. What was the category for the results ranging between 71-74? Needs explanation

3. Grammatical errors. Example: to examined on page 3, to calculate to calculate on page 6, less that 1% on page 8, eSwatini in discussion, to examined in discussion, and others need to be addressed.

4. It is not good to write a scientific paper in an informal way. For example, starting a new sentence with “And” at many sites, “for example” used repetitively and unnecessarily: in the result part, “This is not a surprising finding”, in the discussion part,…

5. You have reported that you did regression to identify factors associated with the outcome variable. But, there is information on how variables were selected, entered and how the model was fitted, how model adequacy and model fitness were checked.

6. Result:

a. You said “Household characteristics such as household wealth were unavailable and therefore not reported” on page 7. Is it important to report its absence if there is no data? If you are sure its absence can affect the result, better to indicate in limitation section.

b. In table 1, the frequency shall be presented not only percent, Parity- why only First pregnancy listed? Does it mean others do not concerned? In the same table, what is mean by “Number of visits for the pregnancy to the facility?” Is it on the data collection date? Also, “Provider used visual aids, Yes = 7.95%”. What about the rest (no? not known?)

Under Validation measures session,

c. The sensitivity of the self-reported exit interviews was classified as high (>90%), good (≥80 and <90%) and low (<80%). Where is the room for =90? Additionally, the procedure of the classification should be indicated in method section. Reference/evidence should be indicated for the base of the cutoff point.

d. You described, “Indicators with low value ROC results were mostly subjective measures and required a certain level of knowledge and understanding about the service (counselling on pregnancy complications and iron side effects” in result part”. This shall be taken to discussion part and written supported with evidence. It seems judgement not result

e. Similarly, a sentence “The low observed prevalence explains why the two indicators poor reporting given that even small deviations from 100% in specificity can lead to extreme overestimation in a survey” shall be taken to discussion part.

f. Table 3 should contain response categories (“Yes”, “no” report)

g. “Measures of agreement” session should have figure or any other that give the detail information on the issue. Nothing is cited

h. Factors associated ….this session should be written clearly in an informative way. The presence of association is a statistical measurement not a true. The nature of your study design does not allow you to say “There was a significantly higher agreement”. Revise and report your result correctly. You may say, indicator agreement is more likely among….

i. Table 5 must contain reports of Adjusted Odds Ratio. It is totally impossible to declare association only with p-value.

j. References are not important in result section. “Stanton et al. 2013” on page 10

7. Discussion

a. The discussion lacks coherence. Try to keep flow of the idea not to disturb the readers. For example, bring the discussion idea and comparison with the previous studies immediately to each result, not discuss by category/group.

b. Table citation is not important in the discussion. You wrote “(see table 3)”.

c. Make your discussion evidence based. Example “….the low agreement may be because the counselling is not provided as it is recommended.” This lacks reference. Discussion shall base on the result and previous findings.

d. You have “Worth to be noted however is that the Assaf study only focused on the counselling aspect of maternal and child care interventions and did not include the observable components of ANC”. So how does it relate to your study? How does the presence or absence of one indicator affect the agreement of other indicators?

e. As to me, the discussion is not satisfactory, particularly on the factors associated. For example, discuss how different approaches to women of different age can affect the result of data collection and then agreement, how language can affect these data collection methods and then agreement, …

f. It is good if you revise “Limitations and study implications” and “Conclusion”. What should be written under these sections should only be from your finding. References might not be important, the information on which you had no data such as recall bias may not be relevant. Write a concrete conclusion answering your title

Reviewer #3: Manuscript ID: PONE-D-22-23821

Topic: Can women’s reports in client exit interviews be used to measure and track the progress of antenatal care quality? Evidence from a facility assessment census in Malawi

Thanks very much for giving me the opportunity to review and give my comments. My thanks also go to the authors of the manuscript for their interest to deal with this interesting topic addressing the issue of “Can women’s reports in client exit interviews be used to measure and track the progress of antenatal care quality? Evidence from a facility assessment census in Malawi” which are the areas that need more research to cross-check, to address, and to come up with convincing shreds of evidence on reports in client exit interviews be used to measure and track the progress of antenatal care quality. Given all information in mind, the title is impressive, and the way the authors synthesize and present the overall write-up is well documented. I have seen the whole document thoroughly and based on that I have only three comments and one question to be addressed by the authors before the manuscript will publish.

Abstract

In the conclusion part, some of the terms are put as acronyms, for instance, DHS and MICS. Please, write the use the whole word. Because, in the abstract section, acronyms/abbreviations alone are not recommended. Please, write the whole word of the acronyms.

The ethical consideration

The ethical consideration has not been documented. Why ethical clearance has not been taken? Please, clearly stated why ethical clearance has not been taken.

In the result and discussion

In Table 1, the variables parity and provider-used visual aids were not well calculated, and a single category was reported. Here my concern is, why a single category was reported? Please, report all categories.

The discussion is poorly discussed. Please, compare the findings with other findings thoroughly and make them strong as possible.

6. PLOS authors have the option to publish the peer review history of their article (what does this mean?). If published, this will include your full peer review and any attached files.

Reviewer #1: No

Reviewer #2: No

Reviewer #3: No

---

## [Author Response · Author response to Decision Letter 0]

17 Jun 2023

Dr. Sohel Saikat

Academic Editor

PLOS ONE

16/06/2023

Dear Dr. Saikat,

Thank you for inviting us to submit a revised draft of our manuscript entitled, " Can women’s reports of client exit interviews be used to measure and track progress of antenatal care quality? Evidence from a facility assessment census in Malawi” to PLOS ONE. We also appreciate the time and effort you and each of the reviewers have dedicated to providing insightful feedback on ways to strengthen our paper. Thus, it is with great pleasure that we resubmit our article for further consideration. We have incorporated changes that reflect the detailed suggestions you have graciously provided. We also hope that our edits and the responses we provide below satisfactorily address all the issues and concerns you and the reviewers have noted.

To facilitate your review of our revisions, the following is a point-by-point response to the questions and comments delivered in your letter dated 10 May, 2023.

Reviewer comments:

Thanks very much, Dr. Sohel Saikat, for giving me the opportunity to review and give my comments. My thanks also go to the authors of the manuscript for their interest to deal with this interesting topic addressing the issue of “Can women’s reports in client exit interviews be used to measure and track the progress of antenatal care quality? Evidence from a facility assessment census in Malawi” which are the areas that need more research to cross-check, to address, and to come up with convincing shreds of evidence on reports in client exit interviews be used to measure and track the progress of antenatal care quality. Given all information in mind, the title is impressive, and the way the authors synthesize and present the figure is also appreciated. However, the tittle and the findings are not inline, which means there are mismatch between the topic and the figures stated in the result part. I have seen the whole document thoroughly and based on that I have some comments, and questions to be addressed by the authors before the manuscript will publish. 

Abstract

The conclusion section of the abstract should be very conclusive, and the recommendation also included under it. Please, include the recommendation within the conclusion. 

Response- thank you for this suggestion. We have revised the abstract to include recommendations.

In the ethical consideration. You stated that, “…. However, we did obtain approval from the National Directorate of Public Health to use the data in our study”. Would you kindly write the approval reference number taken from the stated directorate? Please, put the reference number to increase the credibility of the data to be taken from the stated directorate. 

Response- We have clarified the status of the study. The study uses secondary analysis and we did not have to request ethics approval or consent from participants, as the data is publicly available on the DHS program website (https://dhsprogram.com/Data/) upon request. 

1. In the methods. The methods section was not appropriately organized. Please, revise this part.

Response- We have taken time to revise and reorganize the methods section based on your recommendation. The new organization provides a clearer and more logical flow of information. 

2. In the result and discussion. In Table 1, the variables parity and provider-used visual aids were not well calculated, and a single category was reported. Here my concern is, why a single category was reported? Please, report all categories.

Response- Thank you for bringing up this issue regarding the calculation of variables parity and provider-used visual aids in Table 1. We have revised the table to include all categories and their respective frequencies. 

3. The discussion is poorly discussed. Please, compare the figures with other findings thoroughly and make them strong as possible. 

Response- Thank you for your feedback regarding the discussion section. We have rewritten the entire discussion section to address this concern. In the revised discussion, we have made a conscious effort to compare our results with relevant studies in the field. We have thoroughly examined the figures and incorporated relevant findings from other research to strengthen our arguments. By doing so, we aim to provide a more comprehensive and robust discussion that highlights the significance of our findings in the broader context of existing literature.

4. What are your recommendations? Based on the result and the conclusion drawn from your study, you are obliged to recommend for the concerned stakeholders to tack corrective measures. 

Response- Thank you for your comment. We have incorporated practical guidance for concerned stakeholders to take corrective measures based on our results and the conclusions drawn from our study. 

Our suggested recommendation is that while household survey programmes such as the DHS and MICS are commonly used as data sources for measuring intervention coverage and quality, policy makers should complement such data with more reliable sources like routine data from health information systems, otherwise there is a risk of making ill-informed policy decisions based on inaccurate information due to longer recall.

Reviewer #1

Overall, the authors addressed an essential topic: measuring prenatal care coverage and quality in Malawi. They conducted a descriptive study of the ANC quality indicators obtained by direct observation and women's self-report in existing interviews using data from the 2013-14 Malawi Service Provision Assessment. The introduction is concisely written and lays out the aims, objectives and rationale of the study clearly. The authors did an excellent job at elaborating on the materials and methods section and have provided sufficient detail to allow understanding and replication of the study. The results were interpreted well using different methods and clearly segregated into relevant parts. The discussion is also well written, and cites from other available evidence to put forward a strong argument in the favor of using other data sources than simply DHS/MICS to measure coverage. 

Suggestion 1-However, the authors do leave the room to elaborate more on implications of these findings in real life and for policymakers. They must add a few lines in the conclusion and hopefully sprinkle it throughout the discussion as well, that would solutions do they propose based on these findings. The authors point a valid problem but it would be ideal if they gave a clear answer to it as well; if not DHS, then what data source can be relied on in most scenarios to estimate coverage, given the resource limitations in LMICs?

Response- Thank you for your comment. We have taken your suggestion into account and have included recommendations in both the limitation and the conclusion section of our study. By doing so, we aim to provide practical guidance for concerned stakeholders to take corrective measures based on our results and the conclusions drawn from our study.

Our suggested recommendation is that while household survey programmes such as the DHS and MICS are commonly used as data sources for measuring intervention coverage and quality, policy makers should complement such data with more reliable sources like routine data from health information systems, otherwise there is a risk of making ill-informed policy decisions based on inaccurate information due to longer recall.

Suggestion 2-Lastly, the authors have addressed all the comments made by previous reviewers, to the extent of changing their analytical approach and rewriting this paper. This is a commendable effort and should be rewarded by publishing with minor revisions (regarding addition of clear policy implications and proposed solutions).

Response- We are sincerely grateful for the recognition of our efforts. We have added clear policy implications and proposed solutions in line with your recommendations. Once again, thank you for your valuable feedback.

Reviewer #2

Dear editor,

Thank you so much for inviting me to assess the scientific quality of the paper titled: “Can women’s reports in client exit interviews be used to measure and track progress of antenatal care quality? Evidence from a facility assessment census in Malawi” Manuscript Number: PONE-D-22-23821

Dear authors,

Thank you for addressing an important area maternal health. I suggest you to clarify the following points in your manuscript.

1. It is not good to call a researchers’ name when using their article. Example Franco et al. on page 3, Bessinger and Bertrand. Better, McCarthy et al, Assaf et al on page 3, and throughout the discussion part. Better to say, “evidence shows, article indicates,…

Response- Thank you for bringing this to our attention. We have removed researchers’ names when referencing their articles throughout the paper.

2. Data and methods: The score was dichotomized into low (≤70 points as 0) and high agreement (≥ 75 points as 1) which was then used to conduct bivariate analysis using the score as the dependent variable on page 6. What was the category for the results ranging between 71-74? Needs explanation

Response- Thank you for bringing this point and allowing us a chance to clarify. Based on the literature, we defined the PABAK values as follows: almost perfect (>0.80), substantial (0.61–0.80), moderate (0.41–0.60), fair (0.21–0.40), slight (0.00–0.20) and poor (<0.00). In this study, four indicators were in the categories of almost perfect to moderate. Based on this, 2 were below 75 and the other were 75 and above. We now have categories for all possible PABAK values and all other measures we use. We appreciate your attention to detail and apologize for any confusion.

3. Grammatical errors. Example: to examined on page 3, to calculate to calculate on page 6, less that 1% on page 8, eSwatini in discussion, to examined in discussion, and others need to be addressed.

Response-We sincerely appreciate your attention to detail and for pointing out the grammatical errors in our manuscript. We acknowledge the presence of errors such errors, we have therefore taken time to thoroughly review the entire manuscript to identify and rectify these grammatical errors to ensure the clarity and accuracy of our work. Thank you once again for bringing these issues to our attention.

4. It is not good to write a scientific paper in an informal way. For example, starting a new sentence with “And” at many sites, “for example” used repetitively and unnecessarily: in the result part, “This is not a surprising finding”, in the discussion part,…

Response- Thank you for your valuable feedback regarding the informal writing style observed in the manuscript. We have updated the style of our writing where applicable.

5. You have reported that you did regression to identify factors associated with the outcome variable. But, there is information on how variables were selected, entered and how the model was fitted, how model adequacy and model fitness were checked.

Response- Thank you for pointing out the discrepancy in our wording. We apologize for the confusion caused. Upon reviewing the methodology section and the data again, we realised that we did not perform a regression analysis as earlier miscommunicated. Instead, we conducted a bivariate analysis, comparing our findings across categories using the chi-square test. We do not use regressions because of the limited number of variables in both datasets. We have corrected the description in the methods section to accurately reflect the methodology employed. 

6. Result:

a. You said “Household characteristics such as household wealth were unavailable and therefore not reported” on page 7. Is it important to report its absence if there is no data? If you are sure its absence can affect the result, better to indicate in limitation section.

Response- Yes, you’re right. It is important to acknowledge and report the absence of household characteristics such as household wealth, especially if they could potentially affect the results. We have moved the discussion about this problem to the limitation section. 

b. In table 1, the frequency shall be presented not only percent, Parity- why only First pregnancy listed? Does it mean others do not concerned? In the same table, what is mean by “Number of visits for the pregnancy to the facility?” Is it on the data collection date? Also, “Provider used visual aids, Yes = 7.95%”. What about the rest (no? not known?)

Response-Thank you for your comment. We have included the frequencies as well as all categories for the respective variables. 

c. Under Validation measures session. The sensitivity of the self-reported exit interviews was classified as high (>90%), good (≥80 and <90%) and low (<80%). Where is the room for =90? Additionally, the procedure of the classification should be indicated in method section. Reference/evidence should be indicated for the base of the cut-off point.

Response- Thank you for your comment. We now make it clear that 90% belongs to the highest category. Additionally, we have included detailed descriptions of the classification procedure for sensitivity and specificity in the methods section of the study, referencing prior studies to support our decisions. You described, “Indicators with low value ROC results were mostly subjective measures and required a certain level of knowledge and understanding about the service (counselling on pregnancy complications and iron side effects” in result part”. This shall be taken to discussion part and written supported with evidence. It seems judgement not result

Response- Thank you for your feedback. Based on your suggestion, we have revised the manuscript, moving the paragraph to the discussion section.

d. Similarly, a sentence “The low observed prevalence explains why the two indicators poor reporting given that even small deviations from 100% in specificity can lead to extreme overestimation in a survey” shall be taken to discussion part.

Response- Thank you for your feedback. Based on your suggestion, we have revised the manuscript, moving the paragraph to the discussion section.

e. Table 3 should contain response categories (“Yes”, “no” report)

Response- Thank you for bringing up this suggestion. We appreciate your input. However, we have decided not to include response categories (“Yes”, “No” report) in Table 3. For the purposes of specificity and sensitivity calculations, our primary interest was not in the response categories themselves, but rather in determining the reported prevalence of those who responded “yes”. We aimed to assess the accuracy of positive responses and did not focus on the explicit inclusion of response categories in Table 3. Including the response categories in the table may lead to unnecessary redundancy and visual clutter. Nevertheless, we have ensured that the presentation and interpretation of the data in Table 3 are clear and understandable for readers.

f. “Measures of agreement” session should have figure or any other that give the detail information on the issue. Nothing is cited

Response- Thank you for pointing out this concern. In the “Measures of agreement “ section, we have included Table 4 to present detailed information regarding the measures of agreement. We have made a reference to Table 4 in the section to clarify this. 

g. Factors associated ….this session should be written clearly in an informative way. The presence of association is a statistical measurement not a true. The nature of your study design does not allow you to say “There was a significantly higher agreement”. Revise and report your result correctly. You may say, indicator agreement is more likely among….

Response-We agree with your observation. We therefore adapted our wording to reflect that the section just conducts a covariate analysis and uses chi-square tests to compare differences in agreement based on women’s demographic characteristics. We have also indicated this as a limitation for that section and suggested thorough quantitative analysis in future studies.

h. Table 5 must contain reports of Adjusted Odds Ratio. It is totally impossible to declare association only with p-value.

Response- We agree that our study does not conclude on the magnitude of associations. We have therefore changed the wording of the section to emphasise that we only conduct a “bivariate analysis”. The limitation of this study was that we could not do a regression analysis to identify factors that are associated with agreement after controlling for other covariates because we have many unobservables. We are therefore hesitant to conclude on the magnitude of the differences across categories.

i. References are not important in result section. “Stanton et al. 2013” on page 10

Response- Thank you, we agree with this observation and have made changes accordingly.

7. Discussion

a. The discussion lacks coherence. Try to keep flow of the idea not to disturb the readers. For example, bring the discussion idea and comparison with the previous studies immediately to each result, not discuss by category/group.

Response- We have restructured and rewrote the discussion section as per your suggestion.

b. Table citation is not important in the discussion. You wrote “(see table 3)”.

Response- We have removed this

c. Make your discussion evidence based. Example “….the low agreement may be because the counselling is not provided as it is recommended.” This lacks reference. Discussion shall base on the result and previous findings.

Response- Thank you for your comment. In the revised version, we have based our discussion on the results obtained in our study and linked them to relevant findings from previous research. Where necessary, we have included appropriate references to support our statements and interpretations.

d. You have “Worth to be noted however is that the Assaf study only focused on the counselling aspect of maternal and child care interventions and did not include the observable components of ANC”. So how does it relate to your study? How does the presence or absence of one indicator affect the agreement of other indicators?

Response- Thank you for your comment. The statement you mentioned has been revised to provide a clearer explanation. The idea was to explain why the findings between the two studies are different. The scope of our study was to assess the validity and reliability of women's self-reports in measuring the quality of antenatal care. Exploring the interplay between different indicators and their agreement in the absence of one specific indicator would require further research and analysis. 

e. As to me, the discussion is not satisfactory, particularly on the factors associated. For example, discuss how different approaches to women of different age can affect the result of data collection and then agreement, how language can affect these data collection methods and then agreement, …

Response- Thank you so much for this suggestion. We agree that addressing these aspects would make that section more convincing. However, we have limited our focus to revising the discussion with the goal to show preliminary linkages between the different social demographic factors and levels of agreement. While these additional factors are not the primary goal of our paper, we have included your suggestion as a potential area of future research.

f. It is good if you revise “Limitations and study implications” and “Conclusion”. What should be written under these sections should only be from your finding. References might not be important, the information on which you had no data such as recall bias may not be relevant. Write a concrete conclusion answering your title

Response- Thank you, the sections have been revised accordingly.

Reviewer #3

Topic: Can women’s reports in client exit interviews be used to measure and track the progress of antenatal care quality? Evidence from a facility assessment census in Malawi. Thanks very much for giving me the opportunity to review and give my comments. My thanks also go to the authors of the manuscript for their interest to deal with this interesting topic addressing the issue of “Can women’s reports in client exit interviews be used to measure and track the progress of antenatal care quality? Evidence from a facility assessment census in Malawi” which are the areas that need more research to cross-check, to address, and to come up with convincing shreds of evidence on reports in client exit interviews be used to measure and track the progress of antenatal care quality. Given all information in mind, the title is impressive, and the way the authors synthesize and present the overall write-up is well documented. I have seen the whole document thoroughly and based on that I have only three comments and one question to be addressed by the authors before. the manuscript will publish.

1. Abstract

In the conclusion part, some of the terms are put as acronyms, for instance, DHS and MICS. Please, write the use the whole word. Because, in the abstract section, acronyms/abbreviations alone are not recommended. Please, write the whole word of the acronyms.

Response- Thank you for taking note of this. We have made revisions accordingly.

2. The ethical consideration has not been documented. Why ethical clearance has not been taken? Please, clearly stated why ethical clearance has not been taken.

Response- Since the study is a secondary analysis, we did not require a request for ethics approval. We have included this statement on ethics approval in the methods section. 

3. In the result and discussion. In Table 1, the variables parity and provider-used visual aids were not well calculated, and a single category was reported. Here my concern is, why a single category was reported? Please, report all categories.

Response- Thank you for this comment. We have included both categories for the two variables. We have also included frequencies for all categories.

4. The discussion is poorly discussed. Please, compare the findings with other findings thoroughly and make them strong as possible.

Response- Thank you for this suggestion. We have revised this section to incorporate findings from the broader literature.

---

## [Decision Letter · Decision Letter 1]

10 Jul 2023

PONE-D-22-23821R1Can women’s reports in client exit interviews be used to measure and track progress of antenatal care services quality? Evidence from a facility assessment census in MalawiPLOS ONE

Dear Dr. Mchenga,

Thank you for submitting your manuscript to PLOS ONE. After careful consideration, we feel that it has merit but does not fully meet PLOS ONE’s publication criteria as it currently stands. Therefore, we invite you to submit a revised version of the manuscript that addresses the points raised during the review process.

**Abstract**

Define what you meant by “objective” and “subjective” measures in the methods section. Were there any cut-off points used?
Include statistical figures in the results section of the abstract. The readers need to know what you meant by “high reporting accuracy” and the validity criteria—same comment for reliability.

**Main body**

        Introduction

One of the study's aims was to uncover factors associated with agreement between women’s self-reports and expert direct observation reports. However, this was not presented in the abstract. Please present statistical summaries in the results section of the abstract.

Methods

Page 6 Variables of Interest: Define the subjective and objective measures (factors) as you used the terms in the abstract. The other option is for the authors to remove the terms from the abstract.

We look forward to receiving your revised manuscript.

Kind regards,

Bereket Yakob, Ph.D.

Academic Editor

PLOS ONE

Journal Requirements:

Reviewers' comments:

Reviewer's Responses to Questions

**Comments to the Author**

1. If the authors have adequately addressed your comments raised in a previous round of review and you feel that this manuscript is now acceptable for publication, you may indicate that here to bypass the “Comments to the Author” section, enter your conflict of interest statement in the “Confidential to Editor” section, and submit your "Accept" recommendation.

Reviewer #2: All comments have been addressed

Reviewer #3: All comments have been addressed

2. Is the manuscript technically sound, and do the data support the conclusions?

Reviewer #2: Yes

Reviewer #3: Yes

3. Has the statistical analysis been performed appropriately and rigorously? 

Reviewer #2: Yes

Reviewer #3: Yes

4. Have the authors made all data underlying the findings in their manuscript fully available?

Reviewer #2: Yes

Reviewer #3: Yes

5. Is the manuscript presented in an intelligible fashion and written in standard English?

Reviewer #2: Yes

Reviewer #3: Yes

6. Review Comments to the Author

Reviewer #2: I have no comment on the current version of the manuscript. As to me , it would be good if published in this form

Reviewer #3: (No Response)

7. PLOS authors have the option to publish the peer review history of their article (what does this mean?). If published, this will include your full peer review and any attached files.

Reviewer #2: No

Reviewer #3: **Yes: **Gossa Fetene Abebe

---

## [Author Response · Author response to Decision Letter 1]

11 Jul 2023

Bereket Yakob, Ph.D.

Academic Editor

PLOS ONE

11/07/2023

Dear Editor

Thank you for inviting us to submit a revised draft of our manuscript entitled, " Can women’s reports of client exit interviews be used to measure and track progress of antenatal care quality? Evidence from a facility assessment census in Malawi” to PLOS ONE. We also appreciate the time and effort you and each of the reviewers have dedicated to providing insightful feedback on ways to strengthen our paper. Thus, it is with great pleasure that we resubmit our article for further consideration. We have incorporated changes that reflect the detailed suggestions you have graciously provided. We also hope that our edits and the responses we provide below satisfactorily address all the issues and concerns you and the reviewers have noted.

To facilitate your review of our revisions, the following is a point-by-point response to the questions and comments delivered in your letter dated 9 July, 2023.

A. Abstract

1. Define what you meant by “objective” and “subjective” measures in the methods section. Were there any cut-off points used?

Response- we acknowledge the potential confusion surrounding the terms "objective" and "subjective" measures, so in our study, we opted to use alternative terminology. Specifically, we employed the terms "observable and concrete measures" to refer to aspects that are externally verifiable and tangible, for example, prescription of medication and "measures related to counselling" to denote variables associated with information sharing between the provider and client, for example, provider counselled the client on iron tablets side effects.

There was no cut-off points used, those words were to describe the interventions

2. include statistical figures in the results section of the abstract. The readers need to know what you meant by “high reporting accuracy” and the validity criteria—same comment for reliability.

Response- Thank you for this suggestion. We have included the statistics in the abstract.

B. Main body

1. Introduction

One of the study's aims was to uncover factors associated with agreement between women’s self-reports and expert direct observation reports. However, this was not presented in the abstract. Please present statistical summaries in the results section of the abstract.

Response- we have rephrased that aim, see below. What we did was to stratify the agreement measure by the social demographic characteristics of the women. Significance was determined by the chi-squared tests. These results were not the main goal and that’s why we did not include the statistics in the abstract but rather list factors that increased the likelihood of reporting reliability based on the chi-squared test and the associated p-value.

Using heterogeneity chi-squared tests, the study also examines whether reporting reliability varies significantly by individual and facility level characteristics.

2. Methods

Page 6 Variables of Interest: Define the subjective and objective measures (factors) as you used the terms in the abstract. The other option is for the authors to remove the terms from the abstract.

Response- We have removed the terms subjective and objective measures from the abstract and throughout the paper to avoid the confusion.

Thank you once again!

Sincerely,

Martina Mchenga, PhD

---

## [Editor Report · Decision Letter 2]

12 Jul 2023

Can women’s reports in client exit interviews be used to measure and track progress of antenatal care services quality? Evidence from a facility assessment census in Malawi

PONE-D-22-23821R2

Dear Dr. Mchenga,

We’re pleased to inform you that your manuscript has been judged scientifically suitable for publication and will be formally accepted for publication once it meets all outstanding technical requirements.

Kind regards,

Bereket Yakob, Ph.D.

Academic Editor

PLOS ONE
---

## [Editor Report · Acceptance letter]

21 Jul 2023

PONE-D-22-23821R2 

Can women’s reports in client exit interviews be used to measure and track progress of antenatal care services quality? Evidence from a facility assessment census in Malawi 

Dear Dr. Mchenga:

I'm pleased to inform you that your manuscript has been deemed suitable for publication in PLOS ONE. Congratulations! Your manuscript is now with our production department. 

Kind regards, 

on behalf of

Dr. Bereket Yakob 

Academic Editor

PLOS ONE